# Systemic Dermatitis Model Mice Exhibit Atrophy of Visceral Adipose Tissue and Increase Stromal Cells via Skin-Derived Inflammatory Cytokines

**DOI:** 10.3390/ijms21093367

**Published:** 2020-05-09

**Authors:** Kento Mizutani, Eri Shirakami, Masako Ichishi, Yoshiaki Matsushima, Ai Umaoka, Karin Okada, Yukie Yamaguchi, Masatoshi Watanabe, Eishin Morita, Keiichi Yamanaka

**Affiliations:** 1Department of Dermatology, Graduate School of Medicine, Mie University, Mie, Tsu 514-8507, Japan; k-mizutani@clin.medic.mie-u.ac (K.M.); white818@med.shimane-u.ac.jp (E.S.); matsushima-y@clin.medi.mie-u.ac (Y.M.); ai618lovemyfamily@yahoo.co.jp (A.U.); okadakarin@clin.medic.mie-u.ac.jp (K.O.); 2Department of Dermatology, Faculty of Medicine, Shimane University, Shimane, Izumo 693-8501, Japan; emorita@med.shimane-u.ac.jp; 3Department of Oncologic Pathology, Graduate School of Medicine, Mie University, Mie, Tsu 514-8507, Japan; masako-i@doc.medic.mie-u.ac.jp (M.I.); mawata@doc.medic.mie-u.ac.jp (M.W.); 4Department of Environmental Immuno-Dermatology, School of Medicine, Yokohama City University Graduate, Yokohama 236-0027, Japan; yui1783@yokohama-cu.ac.jp

**Keywords:** skin inflammation, cytokine, adipose tissue, adiponectin, leptin, thermogenesis

## Abstract

Adipose tissue (AT) is the largest endocrine organ, producing bioactive products called adipocytokines, which regulate several metabolic pathways, especially in inflammatory conditions. On the other hand, there is evidence that chronic inflammatory skin disease is closely associated with vascular sclerotic changes, cardiomegaly, and severe systemic amyloidosis in multiple organs. In psoriasis, a common chronic intractable inflammatory skin disease, several studies have shown that adipokine levels are associated with disease severity. Chronic skin disease is also associated with metabolic syndrome, including abnormal tissue remodeling; however, the mechanism is still unclear. We addressed this problem using keratin 14-specific caspase-1 overexpressing transgenic (KCASP1Tg) mice with severe erosive dermatitis from 8 weeks of age, followed by re-epithelization. The whole body and gonadal white AT (GWAT) weights were decreased. Each adipocyte was large in number, small in size and irregularly shaped; abundant inflammatory cells, including activated CD4+ or CD8+ T cells and toll-like receptor 4/CD11b-positive activated monocytes, infiltrated into the GWAT. We assumed that inflammatory cytokine production in skin lesions was the key factor for this lymphocyte/monocyte activation and AT dysregulation. We tested our hypothesis that the AT in a mouse dermatitis model shows an impaired thermogenesis ability due to systemic inflammation. After exposure to 4 °C, the mRNA expression of the thermogenic gene uncoupling protein 1 in adipocytes was elevated; however, the body temperature of the KCASP1Tg mice decreased rapidly, revealing an impaired thermogenesis ability of the AT due to atrophy. Tumor necrosis factor (TNF)-α, IL-1β and interferon (INF)-γ levels were significantly increased in KCASP1Tg mouse ear skin lesions. To investigate the direct effects of these cytokines, BL/6 wild mice were administered intraperitoneal TNF-α, IL-1β and INF-γ injections, which resulted in small adipocytes with abundant stromal cell infiltration, suggesting those cytokines have a synergistic effect on adipocytes. The systemic dermatitis model mice showed atrophy of AT and increased stromal cells. These findings were reproducible by the intraperitoneal administration of inflammatory cytokines whose production was increased in inflamed skin lesions.

## 1. Introduction

Adipose tissue (AT) is the largest endocrine organ, producing bioactive products called adipocytokines, which regulate several metabolic pathways [1]. AT inflammation induces the dysregulation of adipocytokine production, which plays a critical role in the pathophysiology of atherosclerosis [2]. It is well known that obesity induces AT inflammation and contributes to vascular and AT changes, which may lead to the development of metabolic syndrome. Metabolic syndrome induces insulin resistance that causes hyperglycemia and diabetes, and causes dyslipidemia such as hypertriglyceridemia and low HDL cholesterol, which are risk factors for cardiovascular disorders [3]. In addition, obese AT causes increased angiogenesis, immune cell infiltration, the overproduction of extracellular matrix, and the increased production of pro-inflammatory adipocytokines [4]. As a result, the risks of ischemic heart disease and cerebral strokes increase. 

Chronic inflammatory skin disease is also associated with metabolic syndrome [5,6,7]. However, the mechanism is under discussion, and the contribution of AT is still unclear. We addressed this problem by using keratin 14-specific human caspase-1 overexpressing transgenic (KCASP1Tg) mice, which are accepted as a spontaneous dermatitis model [8]. Caspase-1 is a cytoplasmic cysteine protease characterized by its ability to process the inactive pro-interleukin-1β (IL-1β) into the biologically active IL-1β [9]. IL-1β is an inflammatory cytokine that plays a key role in inflammation and keratinocyte activation. KCASP1Tg mice show severe erosive dermatitis from 8 weeks old, followed by re-epithelization and the development of parakeratotic scale-crust [8]. It was previously reported that a sustained circulating low level of IL-1α/β derived from severe dermatitis causes vascular sclerotic changes, cardiomegaly, and severe systemic amyloidosis in multiple organs [10]. 

In this study, we aimed to investigate the influence of chronic dermatitis in the pathological and functional changes in AT using a KCASP1Tg mouse model.

## 2. Results

### 2.1. GWAT Weight Is Decreased, and Inflammation Affects AT Atrophy

We measured whole-body and GWAT weights with a precision electronic balance to evaluate the effect of dermatitis on the AT. The whole-body and GWAT weights were significantly lower in KCASP1Tg than in wild littermate mice at 10 weeks of age (Figure 1A,B). Next, we evaluated the phenotype of adipocytes in GWAT using hematoxylin and eosin staining (HE). In KCASP1Tg mice, adipocytes were small in size and irregularly shaped compared to in wild type mice (Figure 1C). Abundant mononuclear cell infiltration compared to in control mice suggests the so-called burnout or worn out condition of the adipocytes. We performed immunohistochemical (IHC) staining to identify the stromal cells of AT such as T cells, monocytes and neutrophils. The infiltration of inflammatory cells such as lymphocytes, monocytes and neutrophils was slightly higher in KCASP1Tg mice compared to that in wild type littermates (Figure 1D). 

### 2.2. Activated T Cells and Monocytes Infiltrated the AT

We assessed the number or proportion and characteristic composition of the stromal cells infiltrating the GWAT by flow cytometry. The number of infiltrated stromal cells, including lymphocytes and monocytes, was significantly increased in KCASP1Tg mice (Figure 2A). Next, we verified the characteristics of the lymphocytes and monocytes. We measured the proportion of the CD25+ T cells (Figure 2B). Activated T cells, both CD4+ and CD8+, were increased in KCASP1Tg mice (Figure 2C). Additionally, we measured the proportion of regulatory T cells by using CD127, CD4, CD25 and Foxp3 antibodies (Figure 2D). The regulatory T cells were increased in KCASP1Tg, but there was no significant difference (Figure 2E). Finally, we measured the number of TLR4+ CD11b+ monocytes and proportion of Ly-6C+ CD11b+ monocytes TLR4+ CD11b+ monocytes were significantly increased in KCASP1Tg, and Ly-6C+ CD11b+ monocytes were also increased (Figure 2F,G).

### 2.3. Expression of Inflammation-Related Cytokine Levels in AT

An adipocytokine is a physiologically active substance secreted from adipocytes, and the amount secreted also changes depending on the environment of the adipocytes. Representative adipocytokines such as adiponectin, TNF-α, leptin and MCP-1 were analyzed in the adipocytes of GWAT using a specific ELISA kit (Figure 3). The TNF-α level was significantly elevated in KCASP1Tg mice compared to in controls. Conversely, KCASP1Tg mice showed a lower leptin level.

### 2.4. Changes in UCP1 Expression after Cold Exposure

HSP90, IL-33 and UCP-1 are deeply involved in thermogenesis in adipocytes, and their expression is increased by cold exposure or adrenergic stimulation. We investigated the heat production upon cold exposure challenge and the expression of these thermogenic genes. After 4 °C cold exposure, the mRNA expression of HSP90, IL-33 and UCP1 in adipocytes was quantified using real time-PCR. The mRNA expression of HSP90 and IL-33 was significantly decreased in KCASP1Tg mice (Figure 4A). Although the mRNA expression of UCP1 was significantly increased in KCASP1Tg mice (Figure 4A), the body temperature decreased more rapidly upon exposure to the cold environment, and the KCASP1Tg mice could not survive for a long time (Figure 4B).

### 2.5. Pro-Inflammatory Cytokines Produced in Response to Dermatitis Caused AT Atrophy and Increased Activated Monocyte Infiltration In Vivo

We hypothesized that cytokines produced in inflamed skin affect AT and measured the mRNA expression levels of inflammatory cytokines in the skin of KCASP1Tg and control mice. The mRNA expression levels of TNF-α, IL-1β and IFN-γ in ear skin were measured by real-time PCR. All cytokine levels were significantly increased in KCASP1Tg mice (Figure 5A). To clarify the direct effect of skin-derived cytokines on GWAT, we treated wild-type mice by the intraperitoneal administration of inflammatory cytokines including TNF-α, IL-1β and IFN-γ. The GWATs of the TNF-α-treated wild-type mice were similar to those of KCASP1Tg mice; the adipocytes were large in number, small and irregularly shaped; IL-1β- and IFN-γ-injected mice also showed a similar trend (Figure 5B). The number of infiltrating stromal cells tended to increase in mice that were administered cytokines (Figure 5C); additionally, TLR4+ CD11b+ monocytes were significantly increased in IFN-γ treated mice (Figure 5D).

## 3. Discussion

In the current study, we demonstrated that severe dermatitis induces AT atrophy accompanied by abundant lymphocyte and activated monocyte infiltration, the dysregulation of adipocytokine production and maladaptation to cold environments. We observed not only histopathological AT remodeling but also a GWAT functional change when skin is inflamed. Inflammatory skin disorders are not only skin-related but also involve systemic organ impairment. A previous study showed chronic skin inflammation induces the aberrant remodeling of vascular tissues, potentially resulting in atherosclerosis [11]. Systemic amyloidosis with functional deterioration has also been seen in inflammatory skin conditions [12]. The erupted skin of KCASP1Tg mice produces significantly higher amounts of IL-1α and β compared to normal skin, while the supernatant from skin lesions leads to a severe reduction in lipid particles in an adipocyte culture system [10]. Computed tomography imaging of KCASP1Tg mice at 6 months of age revealed a dramatic decrease in visceral fat compared to in normal controls [10]. Based on these findings, it was hypothesized that persistent dermatitis had a negative effect on visceral AT, and we investigated the state of GWAT using dermatitis model mice.

In KCASP1Tg mice, the whole body and GWAT weights were decreased compared to in wild littermate mice. Each adipocyte was small and irregularly shaped, and showed abundant inflammatory cell infiltration in GWAT. Since increased inflammatory cell infiltration and atrophy of adipocytes were observed, it was considered that AT was strongly inflamed under chronic dermatitis. The stromal cells that infiltrate into GWAT are mainly lymphocytes and monocytes, but it was difficult to quantify them by examination using immunohistochemical staining. We performed stromal cell classification using flow cytometry. It was found that lymphocytes and monocytes, which infiltrated into GWAT, were increased in KCASP1Tg mice. In the lymphocyte fraction, both CD4+ and CD8+ T cells were activated compared to in the control. Moreover, although there was no significant difference, regulatory T cells also tended to increase in the inflamed AT. The CD11b antigen is expressed when premature monocytes are activated to become differentiated monocytes and represents the best molecular marker of the macrophage lineage [13]. In monocytes, the expression of TLR4 is important in mediating inflammatory cytokine production in AT inflammation. In humans, TNF-α suppresses the expression of TLR4 on monocytes; however, IL-6 or IFN-γ mediates the upregulation of TLR4 expression [14]. The number of lymphocytes and TLR4+ CD11b+ activated monocytes was significantly higher in the AT of the KCASP1Tg mice.

AT plays an essential role in metabolic homeostasis by storing and releasing lipids and secreting bioactive proteins and peptides called adipokines, including adiponectin, leptin, MCP-1 and TNF-α in response to environmental changes. Both leptin and adiponectin are mainly produced by adipocytes. The main function of leptin seems to be to provide an afferent signal of nutritional and fat mass status to the hypothalamus, thus controlling energy fat stores and regulating appetite and body weight [15]. Leptin reduces the appetite; increased adipocyte storage leads to higher leptin production, which results in a further loss of appetite. On the other hand, adiponectin helps to repair damaged blood vessel walls, prevent arteriosclerosis, increase insulin’s action and lower blood pressure. However, clinically, blood adiponectin concentration is inversely correlated to visceral fat mass; the mechanism behind this relationship has not been elucidated so far. In psoriasis, one of the most common chronic intractable inflammatory skin diseases, several studies showed that adipokine levels are associated with disease severity [16]. AT plays an important role in the development of psoriasis. Leptin induces IL-6, CXCL-1, IL-8 and MCP-1 production, which is involved in the hyperproliferation of the epidermis in psoriasis [17]. To the contrary, adiponectin is an anti-inflammatory adipocytokine, and when dermatitis is caused in adiponectin knock out mice, the expression of IL-17A, IL-17F and IL-22 increases [18]. Moreover, an increased circulating level of IL-1 has been shown to be accompanied by the local overexpression of TNF-α and IL-12/23 p40 [19], as was also seen in our model mice. However, although our model mice exhibited AT atrophy, psoriasis patients are often obese. BMI is also known to be a risk factor for developing psoriasis [20]. The lower leptin level in KCASP1Tg mice may be responsible for the upregulation of body weight. Additionally, adiponectin production from GWAT was significantly increased because adipocytes oversecrete adiponectin in chronic dermatitis. The effects of systemic dermatitis on AT in our study differ from those in psoriasis and may resemble those in erythroderma, which often results in weight loss. However, a recent report shows that psoriasis patients treated with anti-TNF-α antibodies experience weight gain [21]. If it occurs more often in patients with severe psoriasis and high improvement rates for psoriasis, it is likely that dermatitis has a considerable impact on weight loss.

We speculated that the AT in a dermatitis model shows an impaired thermogenesis ability due to systemic inflammation. After exposure to 4 °C, the mRNA expression of heat-related proteins such as HSP90, IL-33 and UCP1 in GWAT was measured. Heat shock proteins (HSPs) are primary mitigators of cell stress HSPs expression is also induced by cold exposure in brown adipose tissue [22]. It is well known that brown adipocytes play an important role in heat production in mice [23]. When white adipocytes face various stimuli such as cold exposure and adrenergic stimulation, they form beige adipocytes and induce UCP1, producing brown adipocytes [24]. This process can be induced by IL-33, which is released from various cells in response to stress and injury in adult mice and activates type 2 innate lymphocytes (ILC2) [25]. In our study, although IL-33 levels were lower in the KCASP1Tg mice, UCP1 expression was significantly increased after exposure to the 4 °C environment. Therefore, heat production occurred correctly; however, the body temperature of the KCASP1Tg mice decreased more rapidly in the cold environment, and these mice died earlier than the wild-type mice, meaning survival in a cold environment may be harder due to AT atrophy.

We assumed that increased inflammatory cytokine production in the skin lesions was the key factor for this lymphocyte/monocyte activation and AT dysregulation. In fact, TNFα, IL-1β and INF-γ levels were significantly increased in KCASP1Tg mouse ear skin lesions (Figure 5A). To investigate the direct effects of these cytokines, BL/6 wild mice were administrated intraperitoneal injections of TNF-α, IL-1β and IFN-γ, revealing small adipocytes and abundant stromal cell infiltration, suggesting those cytokines exert a synergistic effect on adipocytes (Figure 5B–D). According to our results, inflammatory cytokines whose production is increased by the eczema lesions cause AT atrophy and increased inflammatory cell infiltration in AT. We have reported, in previous studies, that the inhibition of IL-1 can prevent cerebrovascular disease [26]. In this study, we did not prove the prevention of AT damage by treating dermatitis. However, controlling dermatitis with topical corticosteroids, immunosuppressants or anti-cytokine antibodies probably reduce damage to AT. Of course, the long-term use of topical steroids can cause side effects such as skin atrophy, rosacea-like dermatitis and dermatomycosis [27,28], and immunosuppressants or anti-cytokine antibodies may cause severe infections and malignant tumors. We should realize that dermatitis causes damage to various organs and the importance of the active treatment of dermatitis.

There are only a few studies evaluating AT pathology and function in inflammatory skin disease. We found that skin-derived inflammatory cytokines led to the secretion of pro-inflammatory proteins from adipocytes. As a result, activated monocytes and lymphocytes infiltrated into the AT, which lead to AT atrophy and increased the difficulty in the metabolic adaptation to cold temperatures. Our study suggested that inflammatory skin diseases lead to organ damage and require careful attention.

## 4. Materials and Methods

### 4.1. Mouse Study

We used 8–10-week-old spontaneous dermatitis transgenic female KCASP1Tg mice, while female littermate C57BL/6 mice were used as controls. The experimental protocol was approved by the Mie University Board Committee for Animal Care and Use (Protocol #22-39-2, approved date, 1 October 2014). Eight-week-old female wild mice were also administered intraperitoneal injections of recombinant tumor necrosis factor-alpha (TNF-α), recombinant IL-1β, recombinant interferon-gamma (IFN-γ) or phosphate-buffered saline (PBS). The cytokines were diluted in PBS and 250 µg/kg of body weight each time were injected three times per week for two weeks. All the recombinant compounds were purchased from Biolegend (San Diego, CA). All the mice were sacrificed the day after the sixth injection. For the cold stimulation study, 10-week-old KCASP1Tg and littermate mice were housed, one per cage, at a temperature of 22 °C. All the mice were transferred to a 4 °C room, and body temperature was measured every hour using thermal imaging camera FLIR i5 (Extech Instruments, Waltham, MA, USA). After eight hours, all the mice were sacrificed and AT was sampled.

### 4.2. Tissue Sampling

Isolated adipocytes were prepared as previously reported with minor modifications [29,30]. In brief, extracted gonadal white AT (GWAT) was minced and incubated in PBS with 0.5% bovine serum albumin (BSA, Sigma-Aldrich, St. Louis, MO, USA), 10 mM CaCl_2_, and 2 mg/mL of collagenase type Ⅱ (Worthington Biochemical Corporation, Lakewood, NJ, USA) at 37 °C for 20 min at 200 rpm. The suspension was filtered through a 100-µm nylon filter and centrifuged at room temperature at 400 rpm for one minute; the floating cells (adipocytes) were then collected. The remaining suspension was further centrifuged at 1500 rpm for 10 min. The supernatant was decanted and the remaining cell pellets were resuspended with 0.5% BSA/PBS. We counted the number of viable cells based on trypan blue exclusion and diluted the cell suspensions to final concentrations of 5–10 × 10^6^ cells/mL as adipose-infiltrated stromal cells.

### 4.3. Histological and Immunohistochemistry Analysis

Tissue was fixed in 10% formalin neutral buffer solution (Wako, Osaka, Japan), embedded in paraffin, cut into 3.5-µm sections and stained with hematoxylin and eosin (HE). Immunohistochemistry (IHC) was performed with primary antibodies—rabbit anti-mouse CD4 antibody (Cell Signaling Technology, Danvers, MA, USA), rabbit anti-mouse CD8a antibody (Cell Signaling Technology), rabbit anti-Ly6G antibody (Abcam, Cambridge, UK) and rat anti-Ly6C antibody (abcam)—and secondary antibodies—goat anti-rabbit immunoglobulins/biotinylated (Dako, Santa Clara, CA, USA) and simple stain MAX-PO (rat) (Nichirei Biosciences, Tokyo, Japan). The adipocytes were observed with 400× HE and counted, and the area was measured using ImageJ and Adiposoft (NIH, Bethesda, MD, USA).

### 4.4. Flow Cytometry Analysis

AT-infiltrating stromal cells were stained with FITC anti-mouse CD3ε antibody, PE anti-mouse CD4 antibody, PE/Cy7 anti-mouse CD25 antibody, APC anti-mouse CD8a antibody, FITC anti-mouse CD127 antibody, PerCP/Cy5.5 anti-mouse CD4 antibody, APC anti-mouse CD25 antibody (these antibodies were purchased from Biolegend, San Diego, CA, USA), PE anti-mouse/Foxp3 antibody (Invitrogen, Carlsbad, CA, USA), PE anti-mouse CD11b antibody (Biolegend), PE/Cy7 anti-mouse Ly-6C antibody (Biolegend, San Diego, CA, USA) and APC anti-toll-like receptor 4 (TLR4) antibody (Invitrogen, Waltham, MA, USA), and analyzed using an Accuri C6 (Becton, Dickinson and Company, Franklin Lakes, NJ, USA).

### 4.5. Real-Time Polymerase Chain Reaction (PCR)

Total RNA was extracted from ear skin and AT using TRI Reagent (Molecular Research Center, Cincinnati, OH) according to the manufacturer’s protocol. The RNA concentration was measured using a NanoDrop Lite spectrophotometer (Thermo Fisher Scientific, Worsham, MA, USA) and 1 µg of total RNA was converted to cDNA using the High-Capacity RNA-to-cDNA Kit (Applied Biosystems, Foster City, CA, USA). The TaqMan Universal PCR Master Mix II with UNG (Applied Biosystems, Foster City, CA, USA) was used to measure the mRNA expression of TNF-α (Mm00443258_m1), IL-1β (Mm01336189_m1), IFN-γ (Mm01168134_m), heat shock protein 90 (HSP90, Mm00658568_gH), IL-33 (Mm00505403_m1) and uncoupling protein 1 (UCP-1, Mm01244861_m1). All probes were purchased from Applied Biosystems and glyceraldehyde-3-phosphate dehydrogenase (Mm99999915_g1) was used as the internal control. Amplification was performed in a LightCycler 96 System (Roche Diagnostics, Basel, Switzerland). The cycling parameters were as follows: 95 °C for 30 s, followed by 40 cycles of amplification at 95 °C for 5 s and 60 °C for 30 s.

### 4.6. Enzyme-Linked Immunosorbent Assay (ELISA)

The total protein content from adipocytes was extracted using a radioimmunoprecipitation assay buffer (Nacalai tesque, Kyoto, Japan). Adiponectin, TNF-α, leptin and monocyte chemotactic protein-1 (MCP-1) levels were measured using specific ELISA kits (R&D systems Minneapolis, MN) according to the manufacturer’s protocol, and the data were normalized to GWAT weight. The absorbance was measured using MULTISKAN JX (Thermo Fisher Scientific, Worsham, MA, USA). The values were analyzed using Ascent Software for Multiskan Ascent (Thermo Fisher Scientific, Worsham, MA, USA).

### 4.7. Statistical Analysis

Statistical analysis was performed using the PRISM software version 8 (GraphPad, San Diego, CA, USA). The Mann–Whitney U test was used to compare variables between groups. Differences were considered significant at *p* < 0.05.

## 5. Conclusions

The systemic dermatitis model mice showed atrophy of AT and an increased infiltration of activated T cells and monocytes into the AT. These were reproducible by the intraperitoneal administration of inflammatory cytokines such as TNF-α, IL-1β and IFN-g, whose production was increased in inflamed skin lesions. The persistence of skin inflammation can potentially cause negative reactions in AT.

## Figures and Tables

**Figure 1 ijms-21-03367-f001:**
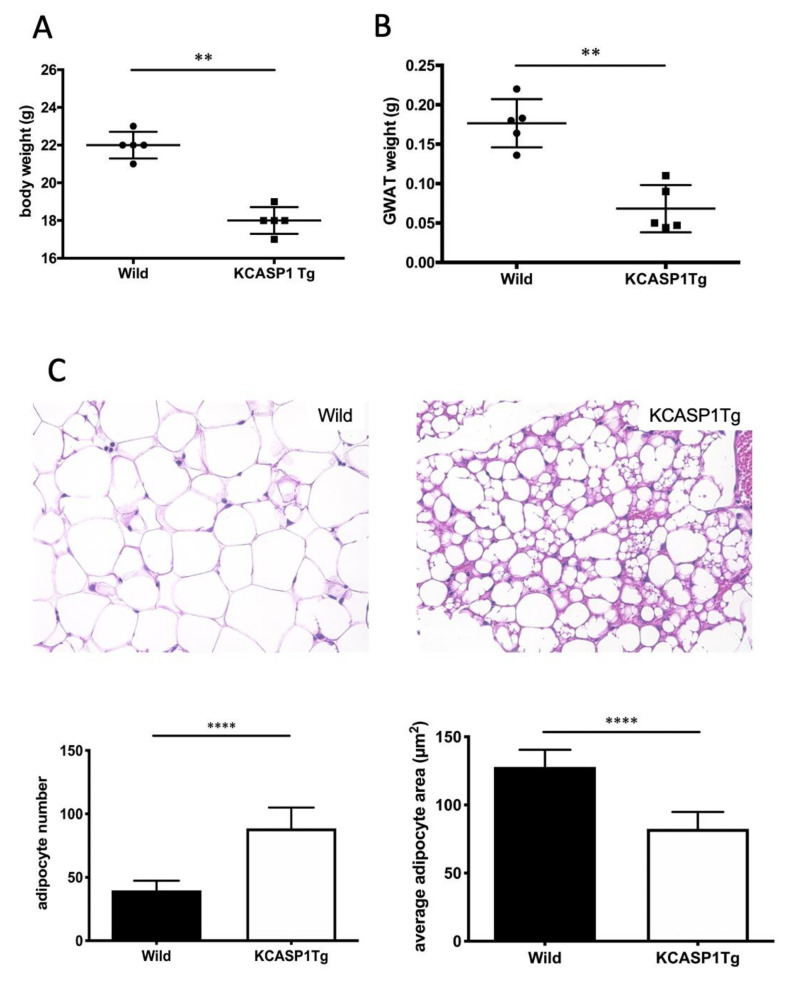
KCASP1Tg mice showed low gonadal white adipose tissue (GWAT) weights and degenerated adipose tissue (AT). (**A**) The whole body and (**B**) GWAT weights were decreased in KCASP1Tg compared to wild littermate mice at 10 weeks of age. (**C**) Representative photomicrographs of the hematoxylin and eosin (HE) (×400) staining of the GWAT of wild littermate and KCASP1Tg mice at 10 weeks of age. Histological sections were 3.5 µm thick. Adipocyte number and adipocyte area were measured by using ImageJ and Adiposoft (three parts of each slide, *n* = 3 in each group). Adipocytes were significantly smaller and larger in number in KCASP1Tg. (**D**) Immunohistochemical staining (×400) was performed to characterize stromal cells. Lymphocytes, monocytes and neutrophils were slightly higher in KCASP1Tg but not significantly so. ** *p* < 0.01, **** *p* < 0.0001 versus control.

**Figure 2 ijms-21-03367-f002:**
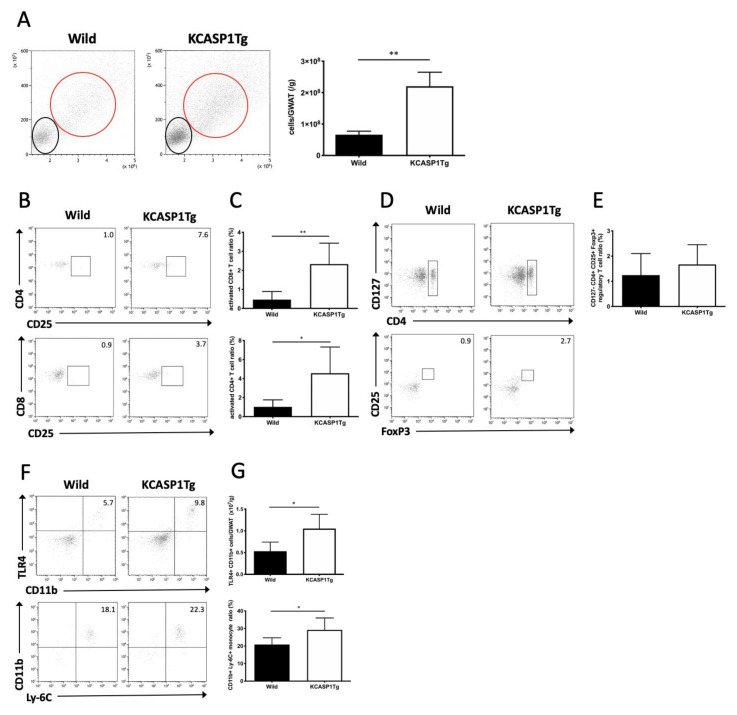
AT cell infiltration in KCASP1Tg mice. (**A**) Flow cytometric analysis of infiltrated stromal cells in GWAT. The black and red circles correspond to the lymphocyte and monocyte populations, respectively, in the left figure. Stromal cell infiltration, including lymphocytes and monocytes, was significantly increased in KCASP1Tg mice. (**B**,**C**) The population of CD25+ activated lymphocytes was measured in CD4+ or CD8+ lymphocytes. Activated T cells were significantly increased in KCASP1Tg. (**D**,**E**) The proportion of regulatory T cells in AT was measured. There was no significant difference between KCASP1Tg and wild mice. (**F**,**G**) The proportion of Ly-6C+ CD11b+ and number of TLR4+ CD11b+ activated monocytes were evaluated and increased in the AT of KCASP1Tg mice. *n* = 5–6 in each group. * *p* < 0.05, ** *p* < 0.01 versus control.

**Figure 3 ijms-21-03367-f003:**
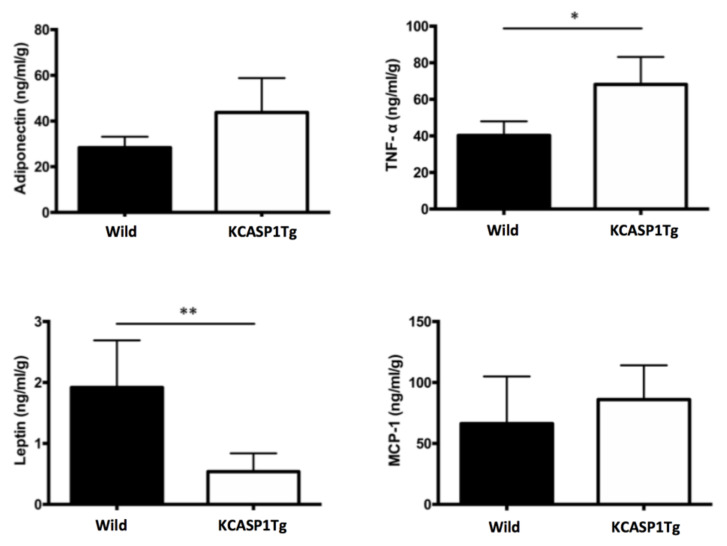
Adipokine expression in AT. We quantified adipocytokines such as adiponectin, TNF-α, leptin and MCP-1 from the adipocytes of GWAT by using a specific ELISA kit. The TNF-α level was elevated in KCASP1Tg mice compared to in controls; conversely, the leptin level was decreased in KCASP1Tg (*n* = 5 in each group). * *p* < 0.05 and ** *p* < 0.01 versus control.

**Figure 4 ijms-21-03367-f004:**
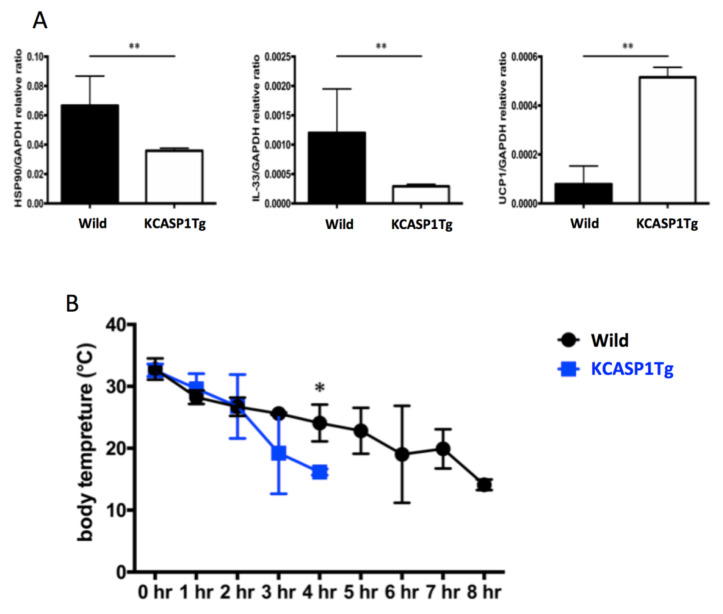
mRNA expression of heat-related proteins in GWAT after cold exposure. (**A**) The mRNA expression of HSP90 and IL-33 was decreased, while that of UCP1 was increased in the adipocytes of KCASP1Tg mice after exposure to 4 °C (*n* = 5, each group). (**B**) The body temperature of mice exposed to 4 °C was measured every 1 hour using thermal imaging camera. The body temperature of the KCASP1Tg mice decreased more rapidly in a cold environment (*n* = 5, each group). * *p* < 0.05, ** *p* < 0.01 versus control.

**Figure 5 ijms-21-03367-f005:**
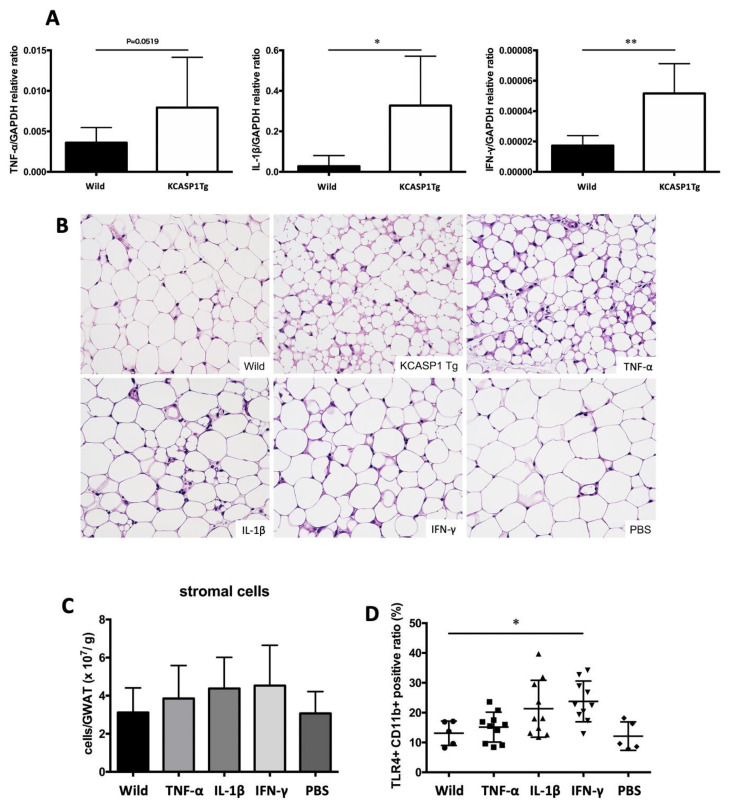
Production of inflammatory cytokines in the skin and the direct effect on AT. (**A**) The TNFα, IL-1β and INF-γ levels were significantly increased in KCASP1Tg mouse ear skin lesions. (**B**) Eight-week-old BL/6 wild mice were treated by intraperitoneal injections of TNF-α, IL-1β, IFN-γ (250 µg/kg body weight/each time) or PBS three times per week for two weeks. The AT of TNF-α-treated mice showed a similar appearance to that of KCASP1Tg mice; IL-1β- and IFN-γ-administrated mice also showed a similar trend; adipocytes were large in number, small and irregularly shaped (HE, ×400). (**C**,**D**) The number of infiltrating cells in GWAT was counted, and the infiltrating cells were analyzed by flowcytometry. The number of infiltrating cells in GWAT was increased in cytokine-administrated mice, while TLR4+ CD11b+ activated monocytes were significantly increased in IFN-γ-treated mice. *n* = 5 per group. * *p* < 0.05, ** *p* < 0.01 versus control.

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
