# Peer review of "Systemic Dermatitis Model Mice Exhibit Atrophy of Visceral Adipose Tissue and Increase Stromal Cells via Skin-Derived Inflammatory Cytokines"

_ijms, 2020, doi:10.3390/ijms21093367_

Round 1
Reviewer 1 Report
In this paper Mizutani and collegues investigated the pathological and functional changes in adipose tissue and gonadal white adipose tissue in a mouse model spontaneous dermatitis.
The manuscript has been strongly improved in comparison to the previously version 1.
Comments: the quality of the figures 1 C and 1 D is very low and it not possible to distinguish the structures and the cells described in the results. Please include more representative results
The title of section 2.5 must be revised
The title of the manuscript must be revised: it is too general and do not well describe the model used
Also in the abstract the last sentence does not clearly explain the results obtained in the paper
Author Response
Responses to the comments of Reviewer #1
In this paper Mizutani and collegues investigated the pathological and functional changes in adipose tissue and gonadal white adipose tissue in a mouse model spontaneous dermatitis. The manuscript has been strongly improved in comparison to the previously version 1.
Comments:
#1 the quality of the figures 1 C and 1 D is very low and it not possible to distinguish the structures and the cells described in the results. Please include more representative results
Response: Thank you for your suggestion. We have replaced the more clear and representative figure.
#2 The title of section 2.5 must be revised
Response: We revised the title of section 2.5.
#3 The title of the manuscript must be revised: it is too general and do not well describe the model used
Response: Thank you for your suggestion. We revised the title of manuscript.
#4 Also in the abstract the last sentence does not clearly explain the results obtained in the paper
Response: We have revised the abstract.
Reviewer 2 Report
1.The new title is bbetter than the first one
2.The supplemented material added to the text is in agreement with the suggestions asked by the first review.
3.Responses about TEWL and microbioma are ok especially the fact
that TEWL is reduced in the presence of visceral obesity,but how did authors explain the hyperhidrosis of the obese patients?What about mices?
4.Now the chapter of Conclusions was introduced and is better than the last sentence of the first submitted variant
6.Decision-Accept the paper
Author Response
Responses to the comments of Reviewer #2
#1 The new title is bbetter than the first one.
Response: Thank you very much.
#2 The supplemented material added to the text is in agreement with the suggestions asked by the first review.
Response: Thank you very much.
#3 Responses about TEWL and microbioma are ok especially the fact that TEWL is reduced in the presence of visceral obesity, but how did authors explain the hyperhidrosis of the obese patients? What about mices?
Response: Thank you for your question. We also thought that TEWL would increase because obesity often causes hyperhidrosis. In fact, TEWL naturally increases in hyperhidrosis patients. However, in obesity patient, the skin temperature in the abdomen is decreased and that of hands tend to increase because adipose tissue inhibits abdominal heat transfer (Savastano DM, et al. Am J Clin Nutr. 2009;90(5):1124‐1131). It is unclear if these are related to TEWL, but we think there are still many points to be explored regarding obesity and skin disease. We will consider as future research. Thank you again for your suggestion.
#4 Now the chapter of Conclusions was introduced and is better than the last sentence of the first submitted variant.
Response: Thank you very much.
#5 Decision-Accept the paper.
Response: Thank you very much.
This manuscript is a resubmission of an earlier submission. The following is a list of the peer review reports and author responses from that submission.
Round 1
Reviewer 1 Report
First remark is related to the title-The adipose tissue is a part of the skin-epidermis,dermis and hypodermis are skin components ,so the authors should find a better title to express the interactions between components of the skin.
Page 1 lines 43-49.Please add informations about high glucose level and blood lipids in metabolic syndrome.
Page 7 lines 190-192 and 196-199.Please comment and explain the fact that you founded AT atrophy with the fact that in metabolic syndrome associated with psoriasis the abdominal fat correlated with the visceral fat is higher and not atrophyc.What do you expect to happens if you use TNF inhibitors as adalimumab in KCASP1Tg mice.
Please comment in discussion part about the other IL levels as IL 17,IL 22/23,IL 6.
Please discuss about the role of topical steroids used for treating inflammatory skin diseases to prevent systemic metabolic or AT changes,also related with the wellknown adverse reactions when they are prolonged used(Topical Steroid Induced Facial Rosaceiform Dermatitis Acta Endo (Buc) 2016 12: 232-233, or Bacillus cereus strain isolated from Demodex folliculorum in patients with topical steroid-induced rosaceiform facial dermatitis.An Bras Dermatol. 2016;91:676-78,)
Did authors tried to measure the sirtuins pathway and its interactions with leptine or adipokines.
What authors suggests about the interactions of skin inflammation with joints directly,or via AT or by citokines or via beta blockers (Immunologic adverse reactions of β-blockers and the skin (Review) . Exp Ther Med.2019;18(2):955-959, Metoprolol-associated onset of psoriatic arthropathy . Am J Ther.2017;24(3);e370-e371)
Did authors find any correlation between trans epidermal water loss and the reduced AT.Please comment.What do authors think about the influence of the skin microbioma and the AT and about the microbioma changes to activate TLR.
Lines 241-242 Please detail your suggestion by references because your study did not totally explain this.
Reviewer 2 Report
Shirakami and co-authors studied the connection between skin and adipose tissue in a mouse model of spontaneous dermatitis. They characterized the adipose tissue weight and inflammation in term of leukocyte infiltration, cytokine production. The most part of the data showed in this manuscript are a deeper characterization of this mouse model. The most interesting data on the inflammatory characteristics of gonadal white adipose tissue are weak and must be strongly improved.
Comments:
figure 1A: the histological analysis of KCASP1Tg mice and C57BL/6 mice GWAT adipocite is not clear. The authors described differences in term of shape and size of adipocites. The authors must quantify the differences and give statistical analysis.
-The differences in term of infiltrating leukocytes must be demontrated with IHC analysis.
Figure 2A: The amount of infiltrating leukocytes are characterized by cytofluorimetric analysis but the results must be improved. A deep characterization with specific markers must be performed.
Figure 1C: How GWAT has been quantified?
Paragraph 3.5 describe the characterization of cytokines production by KCASP1Tg and C57BL/6 mice. This paragraph must be described early as one of the first results. This is true also for the results showed in figure 5.
Paragraph 3.3. The experimental procedure must be better described in the result section.